# FedToken: Tokenized Incentives for Data Contribution in Federated Learning

**Shashi Raj Pandey**[1], **Lam Duc Nguyen**[2], and **Petar Popovski**[1]

[1] Department of Electronic Systems, Aalborg University, Aalborg 9220, Denmark.
[2] Software Systems Research Group, CSIRO Data61 Eveleigh, NSW 2016
{srp, petarp}@es.aau.au, lam.nguyen@data61.csiro.au

## Abstract

Incentives that compensate for the involved costs in the decentralized training of a Federated Learning (FL) model act as a key stimulus for clients' long-term participation. However, it is challenging to convince clients for quality participation in FL due to the absence of: (i) full information on the client's data quality and properties; (ii) the value of client's data contributions; and (iii) the trusted mechanism for monetary incentive offers. This often leads to poor efficiency in training and communication. While several works focus on strategic incentive designs and client selection to overcome this problem, there is a major knowledge gap in terms of an overall design tailored to the foreseen digital economy, including Web 3.0, while simultaneously meeting the learning objectives. To address this gap, we propose a contribution-based tokenized incentive scheme, namely `FedToken`, backed by blockchain technology that ensures fair allocation of tokens amongst the clients that corresponds to the valuation of their data during model training. Leveraging the engineered Shapley-based scheme, we first approximate the contribution of local models during model aggregation, then strategically schedule clients lowering the communication rounds for convergence and anchor ways to allocate *affordable* tokens under a constrained monetary budget. Extensive simulations demonstrate the efficacy of our proposed method.

## 1 Introduction

### 1.1 Background and Motivation

The next generation of internet architecture is said to be backed by Web 3.0, a step further toward a fully decentralized online ecosystem powered by blockchain technologies [1–4]. This novel connectivity paradigm, though still in its infancy, opens up unique ways to monetize decentralized participation and allows clients of heterogeneous profiles to engage in the secure exchange of data, assets, and trust. Recent paradigm of privacy-preserving decentralized training of machine learning models, such as Federated Learning (FL) [5–7], is another complementary enabler that is already offering various applications on top of the existing network architecture via distributed computing collaboration on private, untapped data. Therein, it is important to facilitate appropriate participation for distributed training of FL models to support machine learning applications of interest, for instance, in healthcare, automation, logistics and much more [6, 8, 9]. An efficient incentive design is thus a fundamental stimulus for participation in a collaborative operation over a decentralized network infrastructure.

On that pretext, we begin with a natural question on incentive design to enable FL: How to ensure quality participation with contribution-based incentives that meet the objective of communication

Workshop on Federated Learning: Recent Advances and New Challenges, in Conjunction with NeurIPS 2022 (FL-NeurIPS'22). This workshop does not have official proceedings and this paper is non-archival.

efficiency simultaneously? This leads to investigating: *(i)* ways to quantify the expected contribution of clients' shared model parameters in improving the model performance during aggregation, given constraints on the offered monetary budget; and *(ii)* transparent distribution of incentives as per the contribution, both, in terms of participation and the contribution made in the FL training. Related works (cf. Section-2) do not cover these issues jointly. In this work, we propose an overall design tailored to the foreseen digital economy that simultaneously meets the learning objectives in terms of training as well as communication efficiency. We propose a *contribution-based tokenized incentive scheme* backed by blockchain technology. It ensures a fair allocation of tokens amongst the clients as per the value of contribution with their shared local model parameters during model training.

Our main contributions are as follows:

- **Tokenized Incentives:** We develop tokenized incentives FedToken[1] for clients to train the FL learning models under budget constraints, reflected with *quota*, at the aggregator. We propose a contribution accounting mechanism based on blockchain. The token distribution algorithm is designed to attribute the value of participation in the model aggregation period.

- **Robust System Design and Improved Performance**: The token allocation scheme accounts for the objective of communication efficiency, achieved as a direct consequence of the strategic selection of client's updates during model aggregation in FL. The rewards for participation are shared fairly: we use temporal discounting amongst the clients that transmitted low-quality model updates. This discourages *free-riders*. We incentivize quality updates in a way that is robust to any choice of the local learning algorithm.

- **Implementation Features and Findings:** The activities regarding the exchange and allocation of tokens are immutably recorded into the Blockchain. The contributors can query the relevant information, such as account balance, receiving tokens, or missing contributions. This is the trusted platform that ensures quality participation in federated learning.

## 2   Related Work

**Incentive Mechanism for FL.** Incentive designs to enable efficient and effective participation of clients in training FL models have been a key focus in recent works [10–12, 2]. In line to this, one fundamental research question on incentive design for FL is the value of *data contribution*. Most incentive mechanism design for FL was investigated dominantly in the wireless regime so as to enable participation while optimizing network resources [11, 10, 13, 12], rather than accounting for the impact of individual data contributions.

The authors in [14] showed a contribution-index based on an approximation to the Shapley value [15, 16] to evaluate data value in FL. The authors in [17] presented a novel multidimensional incentive framework for FL. The authors in [18] proposed a game-based incentive mechanism for an FL platform that combines distributed deep learning and crowdsensing technique for big data analytics on mobile clients. In [19], the authors propose a novel incentive governor for an efficient FL in highly heterogeneous and dynamic scenarios considering contribution and reputation. However, these works primarily have two issues. *First*, the scoring mechanism is too subjective and lacks quality evaluation schemes. *Second*, each participant only has a score and can be easily affected by malicious raters. Therefore, in the absence of a trusted mechanism for recording the offered incentives, the solution is not viable for practical usage. Furthermore, despite a plethora of incentive designs to enable FL, it has not been formalized well in the literature, *what exactly are these incentives, anyway*?

**Incentive Mechanism in Blockchained FL.** Feasibility study on the integration of FL with blockchain is explored in several literature [20–23]. One key benefit is blockchain provides a trustworthy platform to realize *tokenization*, which is a process of creating and managing tokens by recording the verifiable title transfer from one user to another. *Tokens*, in principle, then could quantify offered incentives in FL. Native tokens on a blockchain (e.g., BTC[24] on Bitcoin, ETC on Ethereum [25]) can be used to formulate a value system where the tokens represent monetary value or physical assets offered for quality participation in FL training. However, most of the mentioned works are about latency analysis and scalability issues rather than contribution assessment and token allocation for improving learning objectives in FL.

---

[1]https://github.com/lamnd09/FedToken.

The authors in [26] present a blockchain-based *reputation* system to train FL models. In [27], the (mining) rewards are allocated as per the local data size and compute dominance, which doesn't reflect a practical scenario. Moreover, budget constraints are overlooked, let alone the learning performance. In line with this, the authors in [28] provide a comprehensive survey on blockchain-based FL, where the use of blockchain in FL is reviewed for security breaches and honest participation of clients. The work [23], which is close to our research, introduces a tokenized incentive mechanism where tokens are used as a means of paying for the services that provide participants and the training infrastructure. However, the approach of computing individual contributions is different and for a separate context. In addition, it is intuitively to argue that participation only cannot/shouldn't ensure tokens. We address this and consider *contributions* in participation and allocate tokens without compromising the learning objectives. We show that irrespective of participation frequency, only quality updates are considered for the tokenized incentives, and strategic allocation of tokens jointly improves training and communication efficiency in FL.

## 3 Overall System Design

Consider a set of clients $[N] = \{1, 2, \ldots, N\}$ and the training examples held by each client $n \in [N]$ as a collection of $D_n$ input-output pairs $\{x_i, y_i\}_{i=1}^{D_n}$, where, respectively, $x_i \in \mathbb{R}^d$ is an input and $y_i \in \mathbb{R}$. Each client $n$ exchanges updates on computations of their local data set $\mathcal{D}_n$ with the shared edge aggregator to collaboratively train a learning model. In FL, this translates to finding an optimal global model parameter $w \in \mathbb{R}^d$ that minimizes the empirical risk on all of the distributed training data samples, i.e., the clients collaborate to solve the following optimization problem

$$\min_{w \in \mathbb{R}^d} F(w) + \lambda g(w) \ \text{ where } \ F(w) := \sum_{n=1}^{N} \frac{D_n}{D} f_n(w), \tag{1}$$

where $f_n(w)$ captures the cost of prediction with the model parameter $w$; $\lambda$ is a regularizer; and $g(w)$ is a regularization function to control model complexity and offer *generalization*. In practice, the training examples $(x_i, y_i)$ are drawn from $\mathcal{D}_n$, and as such, the total available training samples is $D = \sum_{n=1}^{N} D_n$. Then, we formally define $f_n(w) = \mathbb{E}_{(x_i, y_i) \sim \mathcal{D}_n}[l(w; (x_i, y_i))]$ as the expected loss. Here, $l(w; (x_i, y_i))$ is the measuring loss function for the prediction [7, 5] over data samples $(x_i, y_i), \forall i \in \mathcal{D}_n$.

**Assumption 1** (**General Assumptions**). *The loss function $f_n : \mathbb{R}^d \to \mathbb{R}$ is twice continuously differentiable, $L-$smooth and $\mu-$ strongly convex, that means, we have*

$$\mu I \preceq \nabla^2 f_n(w) \preceq LI, \forall w \in \mathbb{R}^d,$$

*where $I \in \mathbb{R}^{d \times d}$ is the identity matrix and $\nabla^2 f_n(w)$ is the Hessian of $f_n$ in $w$, respectively.*

Following Assumption 1, we solve the learning problem (1) in its dual form as a particular case of Fenchel duality [29, 10]. The separation of the global problem (1) as primal-dual updates, on the one hand, allows clients to control their local computations in improving the quality of local updates and, on the other hand, facilitates the aggregator to evaluate contributions made by the clients to perform efficient incentives allocation during model training. Then, (1) in its dual form considering the availability of overall $D$ data samples, using $\alpha \in \mathbb{R}^D$ as the dual variable mapping to the primal candidate vector and $\phi(\alpha) = \frac{1}{\lambda D} X \alpha$, is defined as

$$\max_{\alpha \in \mathbb{R}^D} \mathcal{F}(\alpha) := \frac{1}{D} \sum_{n=1}^{D} -f_n^*(-\alpha_n) - \lambda g^*(\phi(\alpha)), \tag{2}$$

where $f_n^*$ and $g^*$ are the convex conjugates of $f_n$ and $g$, respectively, and $X \in \mathbb{R}^{d \times D}$ is a matrix with each column representing data points. Then, we get the mapping between the optimal value of dual variable $\alpha^*$ in (2), and the optimal solution of (1) following the first-order optimality conditions [29] as $w(\alpha^*) = \nabla g^*(\phi(\alpha^*))$. Using a shorthand notation for $\phi \in \mathbb{R}^d$ as $\phi(\alpha)$, in the following, we outline the primal-dual solution abusing distributed computation of clients in a systematic manner.

### 3.1 Primal-dual solution approach

We resort solving (2) in a distributed manner, as shown in [29–31], with the iterative exchange of local dual variables $\alpha_{[n]}, \varrho_{[n]}$ obtained at each client $n \in [N]$. Note that both $\alpha_{[n]}, \varrho_{[n]}$ are vectors

in $\mathbb{R}^D$ with non-zero elements for the residing local data points, i.e., for instance, $(\alpha_{[n]})_i = \alpha_i$ if $i \in \mathcal{D}_n$ and 0 otherwise. The selected clients iterates over their local training examples to output $\alpha_{[n]}, \varrho_{[n]}$. Formally, in one single global communication round $t \in \{1, 2, \ldots, \tau\}$, i.e., the interaction between the edge aggregator and the clients, each selected client $n$ solves the following dual problem in the distributed setting:

$$\max_{\varrho_{[n]} \in \mathbb{R}^D} \mathcal{F}_n\big(\varrho_{[n]}; \phi, \alpha_{[n]}\big), \tag{3}$$

where the weight vector $\varrho_{[n]} \in \mathbb{R}^D$ has elements zero for the unavailable data points and reports value only for the coordinates corresponding to local variables $\alpha_{[n]}$, and 0 otherwise; $\mathcal{F}_n(\varrho_{[n]}; \phi, \alpha_{[n]}) = -\frac{1}{N}g^*(\phi(\alpha)) - \langle \frac{1}{D} X_{[n]}^T \nabla g^*(\phi(\alpha)), \varrho_{[n]} \rangle - \frac{\lambda}{2} \|\frac{1}{\lambda D} X_{[n]} \varrho_{[n]}\|^2 - \frac{1}{D} \sum_{i \in \mathcal{D}_n} f_i^*(-(\alpha_{[n]})_i - (\varrho_{[n]})_i)$ is a corresponding quadratic approximation of $\mathcal{F}$ at $(\alpha_{[n]} + \varrho_{[n]})$ with $X_{[n]}$ as a matrix with $i-$th column having data points for $i \in \mathcal{D}_n$, i.e., $(X_{[n]})_i = (X)_i$, and zero-padded otherwise. As mentioned, the selected devices invest local computations in running stochastic gradient descent (SGD) on their local training samples and later share the output, i.e., the obtained local parameters $\Delta \phi_{[n]}^t := \frac{1}{\lambda D} X_{[n]} \varrho_{[n]}^t$, with the edge aggregator for model aggregation. Meanwhile, the local variable gets updated in each global communication as $\alpha_{[n]}^{t+1} = \alpha_{[n]}^t + \nu \varrho_{[n]}^t$, where $\nu \in [0, 1]$ is the tuneable local aggregation parameter often set between $\frac{1}{N}$ and 1.

## 3.2 Contribution-based Aggregation

Once getting the local updates from $N$ clients, a plain model aggregation scheme is Federated Averaging (FedAvg) [5], i.e., we compute the average of received local updates as

$$\phi^{t+1} := \phi^t + \frac{1}{N} \sum_{n=1}^{N} \Delta \phi_{[n]}^t, \tag{4}$$

and pass $\phi^{t+1}$ back to the $[M] \subseteq [N]$ random clients to compute (3), and iteratively as such, thereafter.

We, however, resort to the engineered, cost-efficient Shapley-based scheme [32, 31] that allows approximation of the contribution of local models during model aggregation to lower the overall communication overhead and design tokenize incentives based on that. The edge aggregator performs $\delta$ random permutations over the received dual variables, where the average marginal contribution is evaluated as

$$u_n = \frac{1}{|M|!} \sum_{p \in \mathcal{P}} \left[ V(S_{n,p} \cup \{n\}) - V(S_{n,p}) \right]. \tag{5}$$

Here, $p \in \mathcal{P}$ is the permutation of clients updates, i.e., $\Delta \phi_{[n]}^t, \forall n \in [M]$, with a probability of $\frac{1}{|M|!}$; $V(\cdot)$ is monotone quantifying performance score evaluated on the test dataset with the permuted subset of client updates, which translates to the improvement in global model accuracy and is measured by $\epsilon$; $S_{n,p}$ is a set of preceding clients before $n$ for a permutation $p$ selected for model aggregation – then, all possible orders of clients contribute to the average contribution (5). Furthermore, we assume the availability of test dataset at the aggregation server; having it as such is a common assumption in FL [6, 33], particularly, for achieving model robustness and a fair level of security guarantees. Then, formally, for a test dataset $(x_i, y_i) \sim \mathcal{D}$, $V(\mathcal{D}) = V_{\text{ref}} - \frac{1}{|\mathcal{D}|} \sum_{i=1}^{|\mathcal{D}|} l(w(\alpha); (x_i, y_i))$ such that $\mathcal{F}(\alpha^*) \le l(;) \le V_{\text{ref}}$ with $\mathbb{E}\left[\mathcal{F}(\alpha) - \mathcal{F}(\alpha^*)\right] < \epsilon$. The method of evaluating the marginal contribution of the local models during the aggregation period in (5) primarily ignores the constrained monetary budget at the aggregator. Said the monetary budget can be quantified as *tokens* (see Definition 1), then, in principle, response to all $M$ queries made by the aggregator, i.e., the local variables received from $M$ participating clients, cannot be incentivized with tokens except with a compensation for their participation as participation reward.

Next, while keeping the learning objectives intact, this raises the following two research questions : (i) *how many updates can the aggregator afford with the given budget*, and (ii) *how to fairly allocate the available budget given only a subset of received local updates contribute in improving the model performance?* This also challenges finding comparable contributions, discouraging free-riders, and handling possible attacks during model aggregation. Therein, the tokens allocation scheme has to be *justified* as per the accounted marginal contribution of the local updates and the available monetary constraints at the edge aggregator. Once done, the records of the contribution accounting mechanism are kept on the Blockchain, hence, allowing contributors to make queries about the relevant information during FL training.

**Algorithm 1** FedToken: Contribution-based Tokenized Incentives in FL

---

1: Initialize: Initial global model $\phi^1$, $[\mathcal{Q}] = \emptyset$, $|\mathcal{Q}|$, $\delta$, learning parameters.
2: **for all** global iteration $t \in \{1, 2, \ldots, \tau\}$ **do**
3:    Select[M]: Select $[M]$ clients at random;
4:    **for all** $m \in [M]$ **do**
5:      Get: Dual variables $\Delta\phi^t_{[m]}, \forall m$ solving (3);
6:    **end for**
7:    SetCounter: $c = 0$;
8:    **while** $c < \delta$ **do**
9:      Set: $v_0^c = V(\emptyset)$;
10:     **for all** $m \in [M]$ **do**
11:       **if** $|V(\phi^c) - v_{m-1}^c| < \epsilon$ **then**
12:         Update: $v_m^c = v_{m-1}^c$;
13:       **else**
14:         Set: $v_m^c \leftarrow V(\{p^c[1], \ldots, p^c[m]\})$;
15:       **end if**
16:       Update: $u_{p^c[m]} \leftarrow [\frac{c-1}{c}]u_{p^{c-1}[m]} + \frac{1}{c}[v_m^c - v_{m-1}^c]$;
17:     **end for**
18:     UpdateCounter: $c = c + 1$;
19:    **end while**
20:    Sort[M]: Sort clients in the descending order of $u_{p^c[m]}$;
21:    Aggr[$\mathcal{Q}$]: Select $|\mathcal{Q}|$ clients updates for aggregation;
22:    **for all** $q \in [\mathcal{Q}]$ **do**
23:      TokenAllocation;
24:    **end for**
25:    Update: global variable as $\phi^t$ (4);
26: **end for**

---

We formalize this with the following definition on the constrained monetary budget at the aggregator.

**Definition 1** (**Quota**). *We define quota $[\mathcal{Q}] \subseteq [M]$ as the affordable number of enrolled model updates with the fixed monetary budget $\mathbb{B}$ that satisfies the pre-defined global model accuracy $0 \leq \epsilon \leq 1$ (i.e., $\mathbb{E}\left[\mathcal{F}(\alpha) - \mathcal{F}(\alpha^*)\right] < \epsilon$) upon the exchange of dual updates $\alpha$. Then, given $\mathbb{B}$ that translates into tokens, we allocate tokens as per the proportion of contribution amongst the chosen $[\mathcal{Q}]$. For simplicity, we assume $\mathbb{B}$ equals to the number possible tokens.*

Note that the dynamic adjustment of $|\mathcal{Q}|$ in relation to changing the value of available budget $\mathbb{B}$ is left as a future work; here, we focus our analysis on a fixed $\mathbb{B}$ and constrained quota. Then, with Definition 1, we have the following proposition.

**Proposition 1.** *For a fixed $\mathbb{B}$, assuming the absence of any malicious updates, a larger size of $[\mathcal{Q}] \subseteq [M]$ undervalues the client's contribution as it leads to a lower average marginal valuation of the client local updates, and hence, affects the number of allocated tokens.*

*Proof Sketch.* The proof follows the properties of standard Shapley value [34] derived from the concept of cooperative games while evaluating the marginal contributions in (5). Take a specific permutation instance $p \in \mathcal{P}$, where the score (or the payoffs) with predecessor set of client $n$, i.e., $S_{n,p} = \{m \in [M]|p(m) < p(n)\}$ is $V(S_{n,p})$. Then, following *Linearity* and *Symmetry* properties of $V(S_{n,p})$, we can generalize $V(S_{n,p} \cup \{n\}) \leq V(S_{n,p} \cup \{n\} \cup \{m\}), \forall n, m \in [\mathcal{Q}]$. That means the average marginal contribution gets lowered, whereas the equality holds only if the local updates from $\{m\}$ are invaluable (or malicious). $\square$

In addition, with the increase in the rounds of permutation $\delta$ and fixed an affordable number of available quotas $|\mathcal{Q}|$, the *range* of $V(\mathcal{Q})$ changes; hence, affecting the quantity of allocated tokens. Given the contributions are only *ranked* in order, $\Theta(\log |\mathcal{Q}|)$, is the best proportional fairness, and enjoys a permutation complexity of $\mathcal{O}(M \log M)$ [32]. The details of the overall operation are provided as a pseudo-code in Algorithm 1.

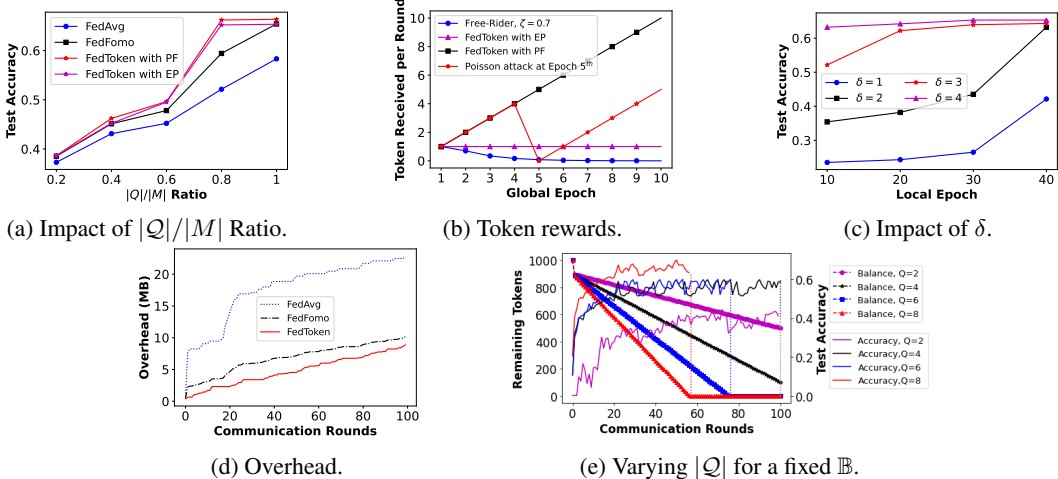

(a) Impact of $|\mathcal{Q}|/|M|$ Ratio.  (b) Token rewards.  (c) Impact of $\delta$.

(d) Overhead.  (e) Varying $|\mathcal{Q}|$ for a fixed $\mathbb{B}$.

Figure 1: Performance Analysis of `FedToken`: a) impact of $|\mathcal{Q}|$ value, b) token Rewards, c) impact of $\delta = \{1, 2, 3, 4\}$ on training performance, d) communication overhead (in MB), and e) varying $|\mathcal{Q}|$.

### 3.3 Tokenized Incentives for Data Contributions

With the primal-dual formalization of the global learning problem, the participating clients can employ any arbitrary algorithm to solve the local learning problem and trade the output dual variables with the edge aggregator. Following the contribution-based aggregation scheme, the edge aggregator efficiently allocates tokens as per its available budget opting for the following simple token distribution schemes: (i) Proportional Fair (PF) allocation, where the tokens are distributed as per the proportion of contribution in improving the global model performance, (ii) Equal Pay (EP), where all participating clients are allocated equal shares of available tokens. Note that the proposed incentive design offers *set-aside* tokens for the initial participation. For $t \in \{2, 3, \ldots \tau\}$, to eliminate the free-riders problem, we multiply the initial allocated slack tokens with a discount factor $\zeta^t < 1$; hence, disincentivizing poor local updates.

## 4 Performance Evaluation

**Dataset and Model.** To validate the effectiveness of our FL design with tokenomic mechanism, we conduct experiments with well-known CIFAR-10 datasets [35], which covers $60000$ data samples, having $32 \times 32$ colored images in 10 classes equally. There are $50000$ training images and $10000$ test images. We leverage a convolutional neural network (CNN) model that is maintained in a distributed manner through blockchain technology. The designed network model possesses hidden layers for feature extraction and fully connected layers for classification.

**Implementation details.** We follow [36] to replicate the non-i.i.d. characteristics of dataset across clients. We set $N = 100$ clients on a single server with 26 core Intel Xeon 2.6 GHz, 256 GB RAM, 4 TB, Nvidia V100 GPU, Ubuntu OS and assume that the clients and server are synchronized, and evaluate the model performance by considering the impact of available quotas as a selection strategy during the model aggregation. For training, each client model is trained for 10 epochs by SGD with a momentum 0.9, batch size of $10$, and the learning rate $0.01$. Unless specified, we set the value of $M = [0.1 \times N]$, which exhibits a good balance between computational efficiency and convergence rate, as demonstrated experimentally in [5]. The default value of discounting factor $\zeta^t$ is set to 0.7.

**Baseline.** The overall system design considers strategic model aggregation for improving the learning objective and distribution of tokens amongst clients for training FL. In particular, the `FedToken` is built up based on the contribution-based aggregation to enhance the quality of updates from FL clients while reducing the communication overhead. Therefore, we evaluate the performance of our proposed `FedToken` with comparable and competitive client selection, and aggregation schemes FedAvg [5], and FedFomo [36], and further present an analysis of two tokenized incentive schemes, as PF and EP. We tokenized the contributions and reward the participating clients based on the fixed monetary budget $\mathbb{B}$ and corresponding affordable quota $|\mathcal{Q}|$ per training round.

**Results**. Fig. 1a shows the impact of available quotas $|\mathcal{Q}|$ on the training efficiency. We see the contribution-based model aggregation strategy strengthens the performance of FedToken, as compared to FedAvg, and FedFomo. Moreover, given a low availability of $|\mathcal{Q}|$, i.e., the monetary constraints or the market for participation, the significance of including quality local updates from FL clients is prominent and is observed to improve the test accuracy up to $12.5\%$ of FL. In specific, when the ratio of $|\mathcal{Q}|/|M| < 0.8$, the FedToken achieves better accuracy and converges faster than the baselines.

Next, in Fig.1b, we provide an evaluation of tokenized incentives per round, where we observe clients are encouraged to join and contribute quality local updates in the training process for higher token offers – leading to competitiveness in individual contributions. The figure compares the amount of tokens received by a client under different scenarios (EP and PF). First, to discourage free-riders and stale updates, as participation tokens are reserved and granted, a discount factor $\zeta^t = 0.7, \forall t$, is multiplied with the allocated tokens. Second, FedToken can effectively detect data poisoning attacks resulting in undesirable local updates, thereby not allocating any participation tokens and saving the token budget for the rest. Fig. 1c demonstrates the impact of random permutation in evaluating the value of local updates in model aggregation. We observe a fundamental limit on imposing larger local computations versus the number of permutations at the aggregator for improving model performance. This means the aggregator can compensate the costs of local computation with added $\delta$, revealing the scope for incentive management.

In Fig. 1d, we analyze the communication overhead of FedToken as compared with the discussed baselines. The proposed approach is communication-efficient and offers better convergence with a low overhead of approx. $72.76\%$ and $29.47\%$ on average in comparison with FedAvg and FedFomo, respectively, due to the strategic selection of local updates as the contribution abiding available quotas. Finally, Fig. 1e evaluates the impact of budget constraint $\mathbb{B} = 1000$ tokens with the affordable quota $|\mathcal{Q}|$ on the learning performance. To begin with, each participant is allocated with a token in their balance and later discounted as per their contribution if not selected during aggregation. We observe a relation between test accuracy and the rounds of communication to exhaust the available tokens for different values of quotas. In principle, strategic selection of clients update allows optimal use of available tokens, as first argued in Fig. 1a. This further opens up several research opportunities, for instance, to design a collaborative data trading market relating tokens and purchasing quotas.

## 5  Conclusion

In this work, we proposed tokenized incentives for clients contributing quality updates to train an FL model in a communication-efficient manner. We developed a holistic approach where token allocation under budget constraints, formalized as quota, is based on the value of contribution during the model aggregation phase, which later favors quality participation by virtue of an efficient selection of local model parameters. The proposed tokenized incentive design enables robust integration of clients with heterogeneous profiles for collaborative FL training while discouraging poor updates and attacks in the foreseen decentralized network architecture as Web 3.0. Finally, we evaluated and analyzed the performance of the proposed approach via extensive simulations.

## 6  Acknowledgements and Disclosure of Funding.

This work was supported by the Villum Investigator Grant "WATER" from the Velux Foundation, Denmark.

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
