# OpenReview forum: "FedToken: Tokenized Incentives for Data Contribution in Federated Learning"
_NeurIPS.cc/2022/Workshop/Federated_Learning — FL-NeurIPS 2022 Poster_

### Official Review · Reviewer_V67a · 2022-10-17

*Summary*:
This work evaluates tokenized incentives for clients in FL where clients are incentivized in proportion to the quality of their local updates with respect to the improvement to the global model. The work proposes to use the Shapley-based scheme to evaluate the contribution of clients' local updates and compensate them accordingly. The work provides evaluation on the CIFAR10 dataset.

*Strength*:
- The paper works on a timely and interesting idea of incentives in FL which is relevant in many scenarios where clients' data contributions are not assumed to be free.
- The paper shows that their proposed method largely improves the quality of the global model while accordingly compensating clients (under the budget) in proportion to their local updates' quality.

*Weakness*:
- The major concern I have is the novelty of the work because it seems very heuristic in that it takes concepts used from Shapley-based schemes in economics and directly uses it in the FL scenario.
- Especially, most of section 3 besides section 3.3 seems to be elaborating on concepts that were previously explored in either the FL or Shapley literature, and I am uncertain of the authors' main contribution.

---

### Official Review · Reviewer_VvRK · 2022-10-18
**Quite uncertain about the dual formulation and proposed approach**

The paper proposes FedToken, an algorithm to incentivize clients for FL using a contribution-based tokenized scheme. To begin, I would like to say that I may not be the best judge for this work as I might be unaware of known results used in this work. As such, there are certain parts of the paper that I did not fully comprehend. Nevertheless, I do have several concerns about the parts that I have tried to understand.

**Concerns.**

* **Assumption 1:** Firstly the assumption should say $\mu$ *strongly* convex and not $\mu$ convex. I am not aware of any works that used the terminology $\mu$ convexity. This itself is quite a strong assumption as the neural networks used in FL training are highly non-convex.

* **Dual formulation in Equation 2:** Equation 2 seems to have multiple typos and/or is simply incorrect. $f_n()$ (and consequently ${f_n}^*()$) is defined for $N$ clients but the summation index in (2) is summing over $D$ ${f_n}^*()$'s. It is not clear what $\alpha_i$ is in the argument to the ${f_n}^*()$ as the index $i$ is not defined. The dimension of $X$ matrix don't match the dimension of $\alpha$. The bigger problem however, is that I'm not sure how the authors have derived this dual formulation. This looks similar to Fenchel duality but still it is not immediately clear how (2) follows from (1). At the very least, the $\alpha_i$' s should be vectors and $\alpha$ should be a matrix. Defining $\alpha$ as a vector in $\mathbb{R}^{D}$ seems surprising to me.

* **Primal-Dual Approach:** It is not clear what the notation $\alpha_{[n]}$ signifies. Does it mean the first $n$ coordinates of $\alpha$? The authors then define a new dual problem in equation (3) and line 131. Again I couldn't follow this derivation.

* **Contribution Based Aggregation:** Several parts in this section were unclear to me. The authors seem to assume that the central server has access to a test dataset without explicitly stating so. It is not clear what "set of preceding clients" means in the definition of $S_{n,p}$ in line 148. Line 153 starts talking about "queries" and "tokens" without describing what they are. Honestly, I found it quite hard to follow the paper from this part onwards.

**General Comments.**

I do believe that the authors are trying to solve a relevant problem with their proposed algorithm. However, I feel the methodology and writing needs a major revision in order to be more accessible to the general public. At the very least, the authors should include detailed proofs about all the derivations used in their work even if they are standard results.

---

### Decision · Program_Chairs · 2022-10-20

Accept (Poster)